

# Spatial and temporal activity patterns of Golden takin (*Budorcas taxicolor bedfordi*) recorded by camera trapping

Jia Li[1], Yadong Xue[2], Yu Zhang[2], Wei Dong[3], Guoyu Shan[3], Ruiqian Sun[3], Charlotte Hacker[4], Bo Wu[1] and Diqiang Li[2]

[1] Chinese Academy of Forestry, Institute of Desertification Studies, Beijing, China
[2] Chinese Academy of Forestry, Research Institute of Forest Ecology, Environment and Protection, Beijing, China
[3] Changqing National Nature Reserve, Hanzhong, China
[4] Duquesne University, Department of Biological Sciences, Pittsburgh, PA, USA

Corresponding author
Diqiang Li, lidq@caf.ac.cn

## ABSTRACT

Understanding animals' migration, distribution and activity patterns is vital for the development of effective conservation action plans; however, such data for many species are lacking. In this study, we used camera trapping to document the spatial and temporal activity patterns of golden takins (*Budorcas taxicolor bedfordi*) in Changqing National Nature Reserve in the Qinling mountains, China, from April 2014 to October 2017. Our study obtained 3,323 independent detections (from a total of 12,351 detections) during a total camera trapping effort of 93,606 effective camera trap days at 573 sites. Results showed that: (1) the golden takin's utilization distributions showed seasonal variation, with larger utilization distributions during spring and autumn compared to summer and winter; (2) the species was recorded at the highest elevations in July, and lowest elevations in December, with the species moving to higher-elevations in summer, lower-elevations in spring and autumn; (3) during all four seasons, golden takins showed bimodal activity peaks at dawn and dusk, with activity intensity higher in the second peak than the first, and overall low levels of activity recorded from 20:00–06:00; and (4) there were two annual activity peaks, the first being in April and the second in November, with camera capture rate during these two months higher than in other months, and activity levels in spring and autumn higher than in summer and winter. This study is the first application of camera traps to assess the spatial and temporal activity patterns of golden takins at a population level. Our findings suggest that the proposed national park should be designed to include golden takin habitat and that ongoing consistent monitoring efforts will be crucial to mitigating novel and ongoing threats to the species.

# INTRODUCTION

Animal behavior as a discipline seeks to understand how animals perceive their external environment and their relationships with surrounding habitat characteristics (*Alcock, 2001*). The adjustment of behavior is a direct manifestation of an animal's response

to environmental stimuli, forming a certain regularity of expected behavioral patterns (*Marco & Marco, 2003*); however, quantifying the behavior of wild animals is challenging (*Rowcliffe et al., 2014*). For example, behavioral metrics and data may be lacking due to the distribution of animals in remote areas that are difficult for humans to access. In addition, the animals of interest may occur at low densities, actively avoid humans, or may be nocturnal or crepuscular (*O'Connell, Nichols & Karanth, 2011*).

Traditional methods for surveying animal behavior depend on direct observation and tagging (e.g., radio tracking via the attachment of telemetry devices to animals) (*Nathan et al., 2012*). However, these methods have limitations (such as complex terrain, dense vegetation and physical capture) that impede the understanding of behavioral ecology. Direct observations can be extremely labour intensive and weakened by the influence of human presence on focal animals (*Nowak et al., 2014*). Further, only a limited number of species are amenable to direct, field-based observation (*Bridges & Noss, 2011*). Tag-based approaches are invasive and can be applied to only a small sample size, which may not be representative of the population at large (*Guan et al., 2013*; *Rowcliffe et al., 2014*). An alternative method is the placement of sensors within the animals' environment, rather than on the animal itself, as is done with camera trapping. Camera trapping can automatically record images of wildlife using motion sensor detection. This removes the need for an observer *in situ* and does not disturb the species being studied (*Forrester et al., 2016*).

Camera trapping is a non-invasive method, can be unattended in the field for several months, and is well suited for studying nocturnal, crepuscular, rare, and elusive species (*Bowler et al., 2017*; *Agha et al., 2018*). Furthermore, animal behavior recorded by camera traps is typically a cumulative composition of many individual animals, allowing for population level measures. Camera traps have become an increasingly popular tool for examining animal behavior as it provides opportunities to undertake extensive and detailed sampling of a species' behavioral repertoire (*Burton et al., 2015*). Many categories of animal behaviour have been previously studied with camera traps, including reproduction (*Crawford et al., 2019*), dispersal or seasonal migration (*Srivastave & Kumar, 2018*), foraging (*Mengüllüoğlu et al., 2018*), and predation (*Caravaggi et al., 2018*; *Akcali et al., 2019*). Activity patterns are a subset of animal behavior. Animals must divide their time between various behaviors. Thus, the proportion of time spent active is a key metric for understanding fundamental behavioral trade-offs, and is the focus of several strands of behavioral and ecological science (*Halle & Stenseth, 2000*; *Rowcliffe et al., 2014*). Camera trapping provides an effective way for researchers to examine activity patterns by extracting behavioral information from camera data (*Bridges & Noss, 2011*). A number of recent studies have demonstrated the utility of this technique to quantify the activity patterns of target species (*Gerber, Karpanty & Randrianantenaina, 2012*; *Ikeda et al., 2015*; *Xue et al., 2015*; *Bu et al., 2016*; *Frey et al., 2017*; *Blake & Loiselle, 2018*; *Bohm & Hofer, 2018*). Resulting data stemming from this method can provide detailed biological information on seasonal activity patterns throughout the year under natural conditions (*Visscher et al., 2017*).

Takins (*Budorcas taxicolor*) are gregarious bovid herbivores comprised of four subspecies that reside in steep and dense montane regions of central and southeastern China, with two

of the four subspecies extending into Bhutan, northeast India and northern Myanmar (*Wu, 1986*; *Sangay, Rajaratnam & Vernes, 2016*). All four subspecies are listed as Vulnerable by the IUCN red list due to their limited geographic range, over-hunting, deforestation, and habitat loss (*Song, Smith & MacKinnon, 2008*). Over the past few decades, the Chinese government has implemented numerous conservation programs, including the Grain-to-Green program and the Natural Forest Conservation Program (*Li et al., 2013*; *Zhang et al., 2007*), to protect and improve habitats for native wildlife. Most of the known key threats for the species are being mitigated, and populations are beginning to increase (*Yuan & Sun, 2007*). To date, previous behavioral studies of the species have included home range size (25–98 km²; *Guan et al., 2015*), feeding type (grazing on non-woody grasses, forbs and bamboo leaves; *Schaller et al., 1986*; *Zeng et al., 2001a*; *Zeng et al., 2001b*), the time of licking salt (usually occurs at 06:30–08:00 and 19:00–20:00; *Ge & Hu, 1988*), determination of rutting season (occurs in summer and calves are born in winter; *Wang et al., 2006*), diurnal activity rhythms and time budgets (*Zeng & Song, 2001*; *Chen et al., 2007*; *Powell et al., 2013*), and seasonal migrations (found to be along altitudinal gradients; (*Guan et al., 2013*). However, the majority of past takin behavioral studies have monitored over only relatively short time periods (e.g., a single season), and data on seasonal and diel activity patterns are still lacking.

The seasonal and spatial activity patterns of golden takins (*B. t. bedfordi*) have received relatively little attention due to the difficulty accessing locations where they occur. In this study, we conducted an intensive long-term camera trapping effort in Changqing National Nature Reserve to survey the activity patterns of golden takins with the following objectives: (1) elucidate seasonal spatial distribution; (2) determine seasonal migration patterns; and (3) examine daily, seasonal, and annual activity patterns. Our results describe the spatial–temporal activity patterns of the species, which can be used to guide species management.

## MATERIALS & METHODS
### Study sites
The study area is in the Changqing National Nature Reserve (107°25′ to 107°45′E, 33°26′ to 33°43′N). Changqing Reserve was established in 1994, selected as the first checklist of the IUCN green list of protected areas in 2014 (https://www.iucn.org/theme/protected-areas), and further upgraded to a National Park in 2017. The reserve is located on the southern slopes of the Qingling mountains in central China, and serves to provide protection for the giant panda (*Ailuropoda melanoleuca*), golden monkey (*Rhinopithecus roxellarae*), and other species in addition to the golden takin. The diverse habitats present support a large number of wildlife species. Among species present, 10 are listed as Class I state key protected wild animals in China and 52 as class II (*Zhao, Zhang & Dong, 2014*). An isolated subspecies of takins, golden takins reside the Qinling mountains (Appendix 1). The area serves as the northern most range of the species, with approximately 5,000 individuals present in total (Forestry bureau of Shaanxi Province, 2001, unpublished data). Approximately 400 individuals inhabit Changqing National Nature Reserve (*Yuan & Sun, 2007*). The reserve

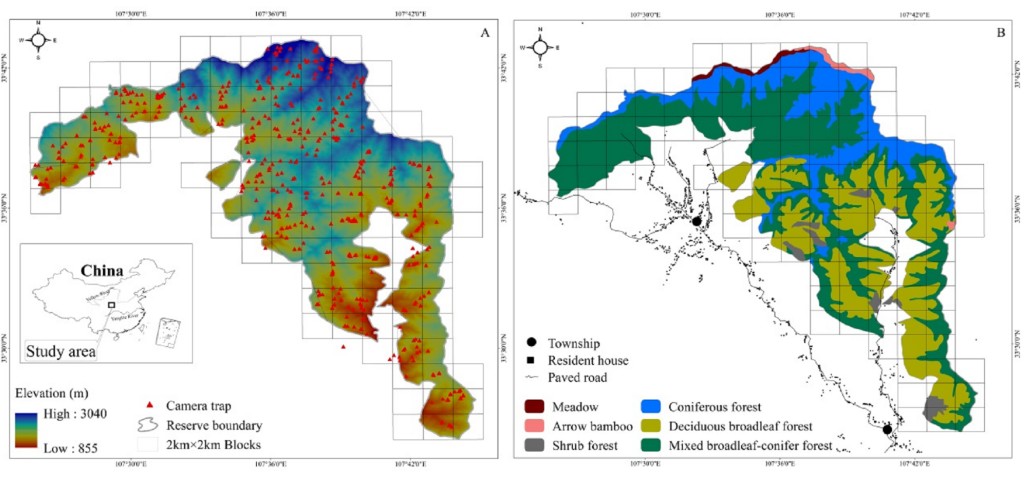

**Figure 1** The Changqing National Nature Reserve is located in central China (A inset). Elevation ranges from about 800 to 3,000 m (A). Forest types are associated with elevation (B). Camera trap locations are indicated by red triangles within 2 ×2 km blocks. .

covers an area of approximately 299 km$^2$, with elevations ranging from 800 to 3,000 m. The average annual temperature is 7 °C and the average annual rainfall is 814 mm. According to local weather data, June through August is the warmest and wettest period, which we termed summer, and December through February is the driest and coldest period, here termed winter, with March through May and September through November forming the seasons of spring and autumn, respectively (*Ren, Yang & Wang, 2002*). Vegetation in the study area varies with elevation (Fig. 1). Deciduous broadleaf forest is mainly found at lower elevation, and the overstory is dominated by oak (*Quercus* spp.), poplar (*Populus* spp.), birch (*Betula* spp.), and Bashania bamboo (*Bashania fargesii*). Shrubs form the understory. Mixed broadleaf-conifer forest is found at mid-elevation, and is dominated by Farges fir (*Abies fargesii*) and Chinese larch (*Larix chinensis*), intermixed with Bashania bamboo and birch. Coniferous forest interspersed with some subalpine shrubs and meadows is found at high elevation. Farges fir, arrow bamboo (*Fargesia spathacea*) and herbaceous plants are common in these zones.

## Data collection

We used 100 infrared cameras (Ltl-6210; Shenzhen Ltl Acorn Electronics Co. Ltd) to survey golden takins in Changqing National Nature Reserve from April 2014 to October 2017. The reserve was divided into 4 km$^2$ blocks using ArcGIS 10.1 (ESRI Inc., Redlands, CA) (total of 118 blocks; cell size 2 km × 2 km, Fig. 1). Each block was then divided into four smaller cells (cell size 1 km × 1 km). We intended on sampling every block but harsh terrain only allowed us to sample only 90 blocks (76% of all 118 blocks in the study area).

We placed one infrared camera in one cell of each block 4 to 6 months, and then moved that camera to an adjacent cell within the same block with > 300 m spacing. Due to harsh terrain leading to difficult navigation in the field, two or more cameras were occasionally placed in one cell. Infrared cameras were placed in areas likely to be used by animals (e.g.,

water holes, trails) and where signs were present (e.g., feces). Thus cameras were distributed across the study area in areas more likely to be used by golden takin. A total of 620 sites were surveyed. Of these 620 sites, 47 failed due to camera damage, malfunction, and loss, resulting in 573 sites for data analysis.

Infrared cameras were attached to trees at an average height of 50 cm above the ground. These were placed 3–8 m away in open habitat clear of vegetation to maximize detection probability and with the aim of obtaining fully body images. Infrared cameras were programmed as follows: mode was set to "Image + Video", the passive infrared sensor was set to moderate, time was set to 24 h a day with a 2-min delay between photographs. Each trigger resulted in 2 photos and a 15 s video, and each photograph was taken with high-quality full colour resolution of 12 Mpx and video at 1080 p resolution. All other default settings were used. Each trigger event automatically collected information on date, time, GPS location and ambient temperature. SD cards and batteries were replaced upon movement of cameras between cells every 4 to 6 months.

## Data analysis

We summarized photographs and videos by location, hour, and date at each camera placement site and considered golden takin detections to be independent if taken at intervals of > 30-min apart (*Blake et al., 2011*; *Li et al., 2010*). The number of effective camera trap days was calculated from the time the camera was placed in operation to the time the last photograph or video was taken if a malfunction occurred (based on date and time stamp). For all analyses we only considered independent detections.

*Spatial analysis*- Kernel home range (KHR) estimates using animal locations are the most commonly used approach to quantify animals' spatial utilization distribution. We used the Animal Movement Analysis Extension (Version 2.0) in Arc View 3.3 (http://www.absc.usgs.gov/glba/gistools/) to determine 50% and 75% (fixed kernel home range method) of camera trapping detected sites to delineate seasonal utilization distribution of all golden takins in Changqing National Nature Reserve. Additionally, we calculated the mean monthly elevation of takins using the number of monthly sites to describe the spatial distribution of the species over time. We interpreted a seasonal change in the spatial distribution of golden takins as evidence of migratory behavior. We used the percentage of independent detections in each vegetation type, deciduous broadleaved forest, mixed broadleaf-conifer forest and coniferous forest, to compare golden takin vegetation use over seasons (Fig. 1).

*Temporal analysis*- We used Capture Rate (*CR*, No. of detections per 100 camera trap days) to estimate the annual activity pattern of golden takins (*Li et al., 2010*; *Blake et al., 2014*). Each camera's independent detections were summed for each month, divided by the number of effective camera trap days for each month, and multiplied by 100 camera trap days, $CR = $ (No. of independent detections / No. of effective camera trap days) $*$ 100 camera trap days. We then calculated *CR* for each camera and each month, and averaged them to estimate a value indicative of annual activity pattern.

All photos recorded date and time. We counted the number of independent detections during each two hour period as an indicator of activity level. We defined two hour time

periods based on seasonal changes in sunrise and sunset: e.g., dawn (06:00–08:00 in spring and autumn, 05:30–07:30 in summer, and 07:30–09:30 in winter), and dusk (16:00–18:00 in spring and autumn, 17:30–19:30 in summer, and 15:30–17:30 in winter), and so on. We used a Daily Activity Index (DAI %) of 2-h durations to examine the daily activity patterns: DAI % = No. of independent detections within each 2-h period / Total no. of independent detections (*Li et al., 2010*).

Finally, we used a One-Way ANOVA to compare elevational change among months and seasons, and capture rate change among months and seasons. This was followed by a Post Hoc Test (Duncan's Multiple Range Comparison Test) to determine significant differences in elevation between pairs of months or seasons. Seasonal migration and activity patterns lend themselves particularly well to graphical representation, and thus the interpretation of these data were based on visual interpretation from boxplots and relative proportions (*O'Connell, Nichols & Karanth, 2011*). Data were expressed as Mean ± Standard Error (Mean ± SE). Statistical significance was considered at $P < 0.05$. Statistical analyses were performed in SPSS 19.0 (IBM Inc., New York, NY).

## RESULTS

### General summary

Takins were recorded at 382 of the 573 camera trap sites. We obtained 3,323 independent detections (out of 12,351 total detections) during a total camera trapping effort of 93,606 effective camera trap days across 573 sites (Table 1). Golden takins were detected at 234 sites (72 blocks) in spring, 115 sites (53 blocks) in summer, 244 sites (69 blocks) in autumn, and 102 sites (48 blocks) in winter. The number of independent golden takin detections fluctuated, with 1,088 independent detections in spring, 1,169 in autumn, and a marked reduction in independent detections over summer (686 independent detections) and winter (380 independent detections).

### Seasonal utilization distribution

Based on visual inspection of the utilization distributions, the golden takin's utilization distribution size varied widely among seasons (Fig. 2). In the spring, the 50% KHR and 75% KHR utilization distributions were calculated as 46 km² and 113 km², respectively. In autumn, the 50% KHR and 75% KHR were 43 km² and 104 km², respectively. Golden takin were mainly distributed in mixed broadleaf-conifer forest (percentage of independent detections, 40.26% and 40.72%, respectively; Table 2 and Fig. 3) and deciduous broadleaf forest (48.35% and 42.00%) in both spring and autumn. The winter 50% KHR and 75% KHR utilization distributions declined to 15 km² and 67 km², with distribution of the species occuring mainly in mixed broadleaf-conifer forest (39.47%), deciduous broadleaf forest (30.79%) and coniferous forest (24.47%). The 50% KHR and 75% KHR in summer were only 12 km² and 29 km², respectively, and were mainly distributed in coniferous forest (42.57%), mixed broadleaf-conifer forest (34.40%), and deciduous broadleaf forest (19.10%).

**Table 1  The number of effective camera days, independent detections, and detections per 100 camera trap days for each month and season for golden takins in Changqing National Nature Reserve.** ANOVA results for differences among months and among seasons are shown at the bottom of their respective columns. Means with different superscript letters are significantly different (P < 0.05) form other means in the same column by Duncnas multiple range test. We defined seasons as spring (March–May), summer (June–August), autumn (September–November), and winter (December–February).

| Season | Month | No. of effective camera days | No. of independent detections | No. of monthly detections per 100 camera trap days (Mean ± SE) | No. of quarterly detections per 100 camera trap days (Mean ± SE) |
|---|---|---|---|---|---|
| Spring | Mar | 6,642 | 193 | (2.77 ± 0.39)[abc] | |
| | Apr | 7,169 | 505 | (6.88 ± 0.91)[e] | (5.13 ± 0.47)[a] |
| | May | 9,412 | 390 | (4.32 ± 0.62)[bc] | |
| Summer | Jun | 9,111 | 261 | (2.51 ± 0.53)[ab] | |
| | Jul | 9,203 | 168 | (1.74 ± 0.33)[a] | (2.23 ± 0.24)[b] |
| | Aug | 8,959 | 257 | (2.41 ± 0.38)[ab] | |
| Autumn | Sep | 8,435 | 387 | (4.66 ± 0.59)[cd] | |
| | Oct | 8,541 | 400 | (4.33 ± 0.44)[bc] | (5.04 ± 0.43)[a] |
| | Nov | 6,844 | 382 | (6.47 ± 1.18)[de] | |
| Winter | Dec | 6,831 | 170 | (2.33 ± 0.39)[ab] | |
| | Jan | 6,740 | 96 | (1.36 ± 0.24)[a] | (1.79 ± 0.18)[b] |
| | Feb | 6,099 | 114 | (1.69 ± 0.29)[a] | |
| Total | | 93,606 | 3,323 | $F = 8.46$, $df = 11$, $P < 0.001$ | $F = 20.73$, $df = 3$, $P < 0.001$ |

## Seasonal migration

Golden takins were detected at elevations ranging from 985 m to 2,958 m and exhibited distinct elevational change across months ($F = 7.31$, $df = 11$, $P < 0.001$) and seasons ($F = 6.23$, $df = 3$, $P < 0.001$). Duncan's multiple range test between months and seasons are shown in Table 3. The species was recorded at the highest elevations in July (2,079 ± 51 m; Table 3) and lowest elevations in December (1,731 ± 56 m). Seasonal migration occurred from May (1,893 ± 32 m) to November (1,766 ± 33 m), as takin steadily ascended in elevation until reaching their highest point in July. Small changes in elevation were also noted between December and February, suggesting a possible migration during this period but that further research is required to determine whether this constitutes a second migration event. From February to April, golden takins gradually returned to lower elevation valleys.

## Daily activity patterns and seasonal differences

Based on visual inspection of the daily activity, the golden takins showed bimodal activity peaks around dawn and dusk during all four seasons (Fig. 4), with activity intensity greater during the second peak (dusk) compared to the first peak (dawn). Lower levels of activity were recorded at night (around 20:00–06:00) relative to daytime. Interestingly, a small number of takins were still moving or engaged in other activities (foraging or mating) later in the night (around 23:30–02:00). In spring and autumn, according to the number of independent detections per two hours, golden takins showed bimodal activity peaks around 06:00–08:00 (DAI = 10.48% in spring and 12.75% in autumn), with the second

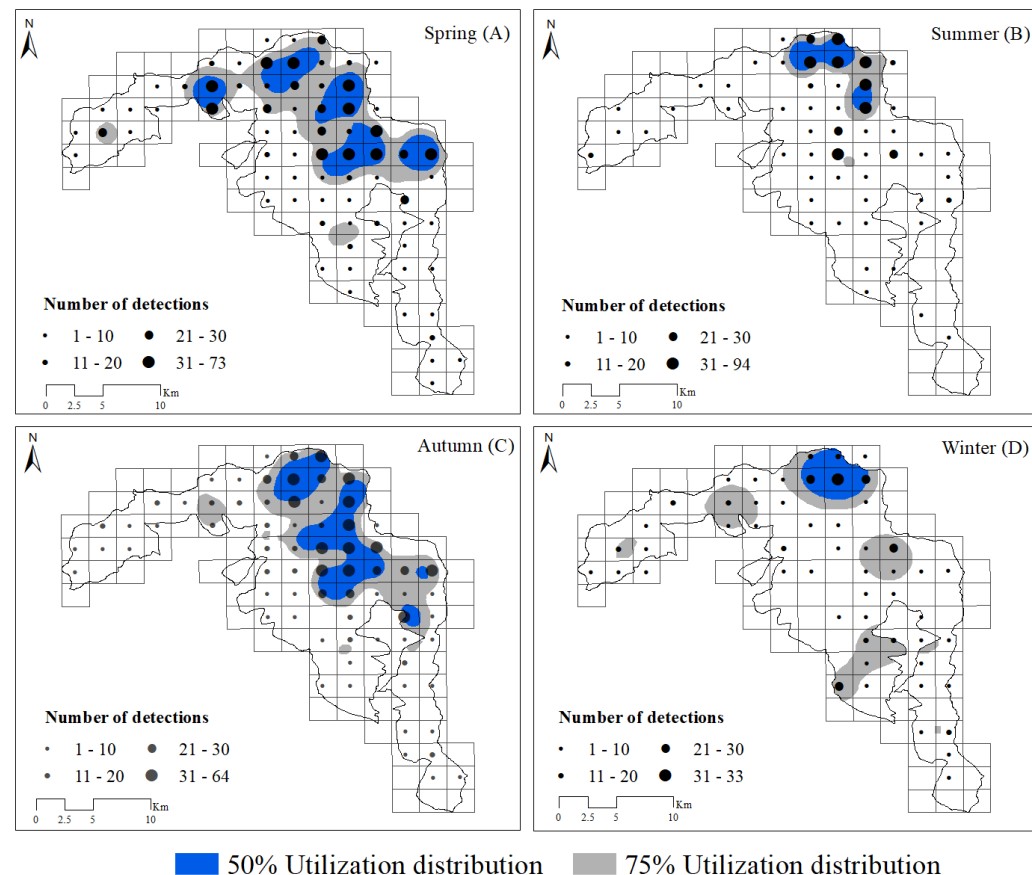

**Figure 2** **Distribution of golden takins in the study area during the spring (A), summer (B), autumn (C) and winter (D).** The black points show the center of each block, and their size indicated the total number of independent camera detections. Note that the numbers indicated by different point sizes differ among seasons. Gray shading indicated the 75% and blue shading indicated the 50% kernel home ranges in each season.

**Table 2** Number and percentage of independent detections of golden takins in different vegetation types across four seasons in Changqing National Nature Reserve.

| Vegetation types | No. of independent detections (%) | | | |
|---|---|---|---|---|
| | Spring | Summer | Autumn | Winter |
| Meadow | 2 (0.18%) | 22 (3.21%) | 3 (0.26%) | 0 (0%) |
| Shrub forest | 3 (0.28%) | 1 (0.15%) | 6 (0.51%) | 1 (0.26%) |
| Arrow bamboo | 6 (0.55%) | 4 (0.58%) | 17 (1.45%) | 19 (5.00%) |
| Deciduous broadleaf forest | 438 (40.26%) | 131 (19.10%) | 476 (40.72%) | 117 (30.79%) |
| Coniferous forest | 113 (10.39%) | 292 (42.57%) | 176 (15.06%) | 93 (24.47%) |
| Mixed broadleaf-conifer forest | 526 (48.35%) | 236 (34.40%) | 491 (42.00%) | 148 (39.47%) |
| Total | 1,088 | 686 | 1,169 | 380 |

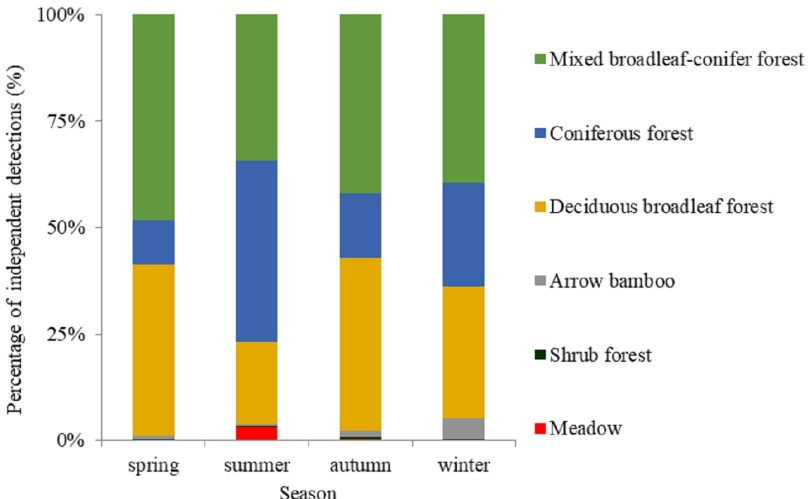

**Figure 3** **The percentage of independent detections of golden takins in different vegetation types across seasons.** Vegetation types in the study area included deciduous broadleaved forest, mixed broadleaf-conifer forest and coniferous forest, shrub forest, meadow, and arrow bamboo.

**Table 3** **The number of sites at which golden takin were recorded by camera traps in Changqing National Nature Reserve during each month and season of the study.** For each month and season, the Mean ± Standard Error elevation (m) is shown. ANOVA results for differences among months and among seasons are shown at the bottom of their respective columns. Means with different superscript letters are significantly different ($P < 0.05$) form other means in the same column by Duncnas multiple range test.

| Season | Month | No. of monthly sites | Mean monthly elevation (Mean ± SE) (m) | No. of quarterly sites | Mean quarterly elevation (Mean ± SE) (m) |
|---|---|---|---|---|---|
| | Mar | 74 | $(1{,}742 \pm 44)^a$ | | |
| Spring | Apr | 146 | $(1{,}753 \pm 24)^a$ | 234 | $(1{,}789 \pm 22)^a$ |
| | May | 97 | $(1{,}893 \pm 32)^{bc}$ | | |
| | Jun | 59 | $(2{,}036 \pm 50)^{de}$ | | |
| Summer | Jul | 50 | $(2{,}079 \pm 51)^e$ | 115 | $(1{,}945 \pm 34)^b$ |
| | Aug | 72 | $(1{,}944 \pm 38)^{cd}$ | | |
| | Sep | 115 | $(1{,}887 \pm 30)^{bc}$ | | |
| Autumn | Oct | 143 | $(1{,}872 \pm 27)^{abc}$ | 244 | $(1{,}806 \pm 22)^a$ |
| | Nov | 118 | $(1{,}766 \pm 33)^{ab}$ | | |
| | Dec | 49 | $(1{,}731 \pm 56)^a$ | | |
| Winter | Jan | 52 | $(1{,}836 \pm 63)^{abc}$ | 102 | $(1{,}755 \pm 41)^a$ |
| | Feb | 60 | $(1{,}771 \pm 56)^{ab}$ | | |
| Total | | 1,035 | $F = 7.31, df = 11,$ $P < 0.001$ | 695 | $F = 6.23, df = 3,$ $P < 0.001$ |

peak in spring around 18:00–20:00 (DAI = 17.19%) and 2-h earlier (16:00–18:00, DAI = 16.85%) in autumn. In summer and winter, golden takins also showed bimodal daily activity patterns with the first peak at 07:30–09:30 in both seasons (DAI = 12.39% in summer and 11.05% in winter), but the second peak in winter around 15:30–17:30 (DAI = 11.05%) and 2-h later (17:30–19:30, DAI =15.45%) in summer.

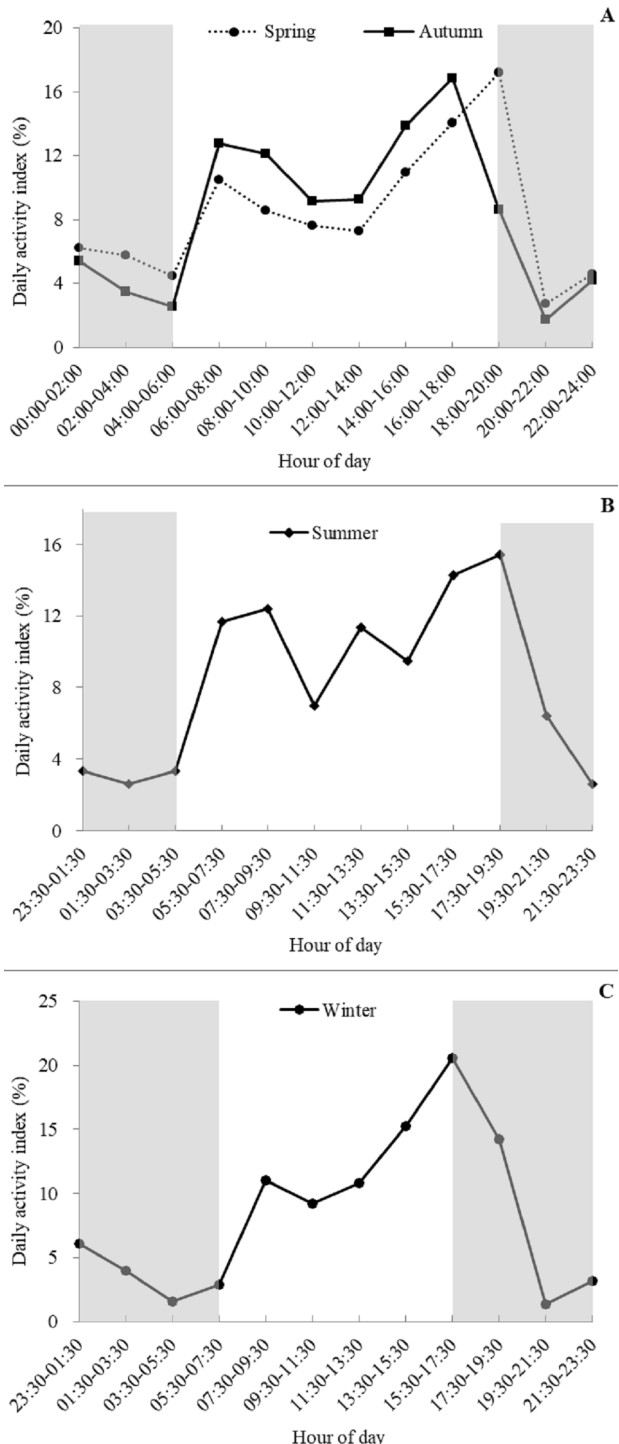

**Figure 4 Seasonal patterns of the daily activity of golden takins.** (A–C) show the mean Daily Activity Index which is the number of independent detection in each 2-h period as a percent of the total number of independent detections. Activity patterns are shown for Spring (A dotted line, circles), Summer (B), Fall (A solid line, squares), and Winter (C). The grey shaded areas indicate night-time hours.

### Annual activity pattern

Capture rates (per 100 camera trap days) differed significantly across months and seasons (Table 1). Activity levels in spring ($CR = 5.13 \pm 0.47$) and autumn ($CR = 5.04 \pm 0.43$) were higher than in summer ($CR = 2.23 \pm 0.24$) and winter ($CR = 1.79 \pm 0.18$) (Table 1). Takins showed: low levels of activity from January to February with a gradual increase in March followed by a peak in April and gradual decline until low levels were reached again in July. Activity then began to increase and reached another peak in November followed again by a gradual decline.

## DISCUSSION

The majority of previous takin activity studies have focused on a single individual's seasonal migration (*Guan et al., 2013*). Information on golden takin daily activity patterns were limited to direct observation (*Zeng et al., 2001a*) or ex situ populations (*Chen et al., 2007*). Our study surveyed golden takins using non-invasive camera trapping continuously from April 2014 to October 2017, covering all four seasons. Our results are the first application of camera traps to elucidate seasonal spatial utilization distribution and altitudinal migration, and the daily, seasonal and annual activity patterns of golden takins at a population level.

Daily activity patterns are often driven by circadian rhythms and periodic changes in environmental stimuli (*Aschoff, 1966*). Animals may optimize time for different activities, and distribute those activities during a 24-h cycle (*Fernandez-Duque, 2003*). Golden takins exhibited bimodal activity peaks at dawn and dusk, similar to previous findings from radio-collared takins (*Zeng & Song, 2001*). Peak activity occurred at dawn and dusk when temperatures were relatively cool with low humidity, and these periods were spent foraging and moving slowly (*Zeng & Song, 2001*; *Chen et al., 2007*). Peaks in rumination behavior often occurred adjacent to peaks in foraging behavior from 10:00 to 16:00 and rumination was often accompanied with resting. Golden takins may lay under tall trees or stone cliffs, overall reducing their movement and energy expenditure during these periods (*Zeng & Song, 2001*; *Chen et al., 2007*). However, some lower levels of activity may still be present. For example, we found that most golden takins rested between 00:00 to 02:00 at night, but some active behaviors were still observed. Previous studies have indicated that there may be a sentry system that exists in some populations during these periods (*Wu, Han & Qu, 1990*).

According to our analysis, the annual activity patterns of golden takins are consistent with seasonal altitudinal migration and reproductive periods. Golden takins were recorded at a lower frequency during calf-rearing months (February to March). During this time, females spend time with their young, and males stay nearby to assist (*Wang et al., 2005*; *Wang et al., 2006*). From June to August, when golden takins were observed in larger groups at higher altitudes and engaging in rutting behavior, the utilization distribution of activities was lower (*Guan et al., 2012*). During migration months (from mid-April to early June), the activities of golden takins were quite evenly distributed and capture rates increased again in October to November, which could be related to a greater effort needed to forage (*Zeng et al., 2010*). Seasonal utilization distribution of golden takins are also consistent
with seasonal migration, as spring and autumn showed larger kernal home ranges than summer and winter.

Seasonal migration is a common survival strategy that allows grazing ungulates to optimize living conditions throughout the year (*Gwynne & Bell, 1968*; *Pettorelli et al., 2007*). We found that seasonal migration patterns of golden takins from high-elevation meadows in summer to mid-elevation fir forest and bamboo in winter, and low-elevation valleys in spring and autumn. This migration pattern was similar to a previously completed study (*Zeng et al., 2008*). Seasonal temperature shifts and changes in plant phenology and its influence on food resources are the likely key forces driving golden takin migration (*Zeng et al., 2001b*). Golden takins gradually descended to lower-elevation presumbly in response to the first greening of vegetation within the valley to replenish energy after the cold winter. The species next ascended to higher-elevations in summer as more nutritious foraging became available. In addition, the cooler environment afforded at high-elevations brings about fewer biting insects and safer mating places (*Wu, Han & Qu, 1990*; *Zeng et al., 2010*). Colder weather, and the senescence of vegetation in high-elevation areas would then force the species to return to lower-elevation ranges in autumn in search of unwithered forage to accumulate energy for the coming winter. However, the golden takin's seasonal migration from autumn to winter was not consistent with observed temperature changes. This could be due to the availability of bamboo and fir forest to use the bamboo at mid-elevations, and the forest acting as a thermal shelter from heavy snow and cold wind. In addition, selection for mid-elevation areas with high solar radiation could be a strategy of thermal adaptation during winter (*Zeng et al., 2008*; *Zeng et al., 2010*; *Guan et al., 2012*). The underlying mechanisms and signals that initiate seasonal migrations are not yet fully understood and require further study.

As a flagship species in China, golden takins are a heavy focus of conservation efforts. The government of China has listed golden takins in the key program of biodiversity conservation (*Ministry of Environmental Protection, 2011*), and has established a national park intended to protect the species in the Qinling mountain range (*State Forestry and Grassland Administration, 2019*). Previous studies have documented that golden takins only reside at elevations higher than 1,360 m in Qinling mountain (*Wu, Han & Qu, 1990*; *Zeng et al., 2008*). However, our camera traps detected a lower elevation for golden takin's distribution (985 m vs. 1,360 m). This change may be due to the implementation of conservation programs that have returned low-elevation farmland to forests. Consequently, we propose to further investigate the golden takins' migration routes in low-elevation areas and call for the inclusion of these elevational ranges in the proposed national park plans.

Human presence and lower quality habitat will most likely create conservation challenges for golden takins (*Zeng et al., 2008*). While it is unclear if human activities affect the species' seasonal migration, the daily life of local villagers is closely linked to reserve resources (e.g., collecting bamboo, and harvesting mushrooms and herbs). Golden takins may also forage and damage local agricultural crops (e.g., wheat, lettuce) in low-elevation areas, where local villagers keep domestic dogs (*Canis lupus familiaris*), creating potential sources of conflicts between humans and takins. Our camera traps detected domestic dogs frequently (20 independent detections and 12 sites) in the forest, indicating that predation pressure on

golden takin calves could be present. Consequently, we recommended that management programs focus on regulating human behavior to avoid conflicts (e.g., tying domestic dogs, changing crop types). In addition, diseases such as conjunctivitus (pinkeye) and tumour growth were first seen on a video of golden takins capture in this study (Appendix 1), showcasing the need for epidemeological studies to be carried out with the species. These many knowledge gaps and potential risks to species preservation warrant the establishment of long-term monitoring programs that regularly assess the advancement of ongoing threats to the species.

Our study had some limitations. First, we were unable to sample some locations due to the difficultly of navigating to remote areas of the study site. This may result in detection error or bias, whereby golden takin may have been present at these sites regardless of our ability to survey them. Second, cameras may miss animals that are present in the study area. We did not test detection probability in this study. However, our results are consistent with previous studies and methods not subject to detection error. In addition, our field survey still covered 76% of the study area, a sizable portion of the intended land mass, and our activity pattern results were consistent with past tagging methods (*Zeng & Song, 2001*; *Zeng et al., 2008*) not subject to detection error. On the other hand, detection probability aggregated across an array of cameras may increase the rate of double detection, leading to inaccurate conclusions surrounding population abundance, behavior (e.g., habitat selection, migration routes, home range, and inter-specific interactions) and distances between camera sightings (*Ikeda et al., 2016*). Double counting, if it occurred, would not be of concern in this study as we focused on questions surrounding activity as opposed to abundance. Accounting for imperfect detection in camera traps surveys of unmarked species is difficult, and identifying individuals, sex and age classes from camera trap images can be challenging. We submit that a combination of techniques such as telemetry and direct observations will be required to supplement studies susceptible to detection error.

## CONCLUSIONS

Our study systematically surveyed golden takin activity across all seasons using infrared camera trapping. Results herein represent the first application of camera traps to elucidate seasonal spatial utilization distribution and altitudinal migration, as well as the daily, seasonal and annual activity patterns of golden takins at a population level. Our findings suggest that the proposed national park should be designed to include golden takin habitat extending to elevations as low as 985 m and that ongoing consistent monitoring efforts will be crucial to mitigating novel and ongoing threats to the species.

## ACKNOWLEDGEMENTS

We would like to thank the Changqing reserve for mangement support. We would also like to thank all staff member of the Changqing Nature Reserve for helping to set camera trpas in the study area. We also thank Yuguang Zhang and Li Yang for their valuable comments for this manuscript, and the two anoymous reviewers for their insightful comments.

### Funding

This work was supported by Nature Reserves Biological Specimen Resources Sharing Sub-platform (No. 2005DKA21404). The funders had no role in study design, data collection and analysis, decision to publish, or preparation of the manuscript.

### Grant Disclosures

The following grant information was disclosed by the authors:
Nature Reserves Biological Specimen Resources Sharing Sub-platform: 2005DKA21404.

### Competing Interests

Wei Dong, Guoyu Shan and Ruiqian Sun are employed of Chang national nature reserve. The authors declare there are no competing interests.

### Author Contributions

- Jia Li conceived and designed the experiments, performed the experiments, analyzed the data, prepared figures and/or tables, authored or reviewed drafts of the paper, and approved the final draft.
- Yadong Xue and Yu Zhang conceived and designed the experiments, performed the experiments, analyzed the data, prepared figures and/or tables, and approved the final draft.
- Wei Dong, Guoyu Shan and Ruiqian Sun performed the experiments, authored or reviewed drafts of the paper, and approved the final draft.
- Charlotte Hacker and Bo Wu analyzed the data, prepared figures and/or tables, and approved the final draft.
- Diqiang Li conceived and designed the experiments, analyzed the data, authored or reviewed drafts of the paper, and approved the final draft.

### Data Availability

Raw data is available in the Supplemental Files.

### Supplemental Information

Supplemental information for this article can be found online at http://dx.doi.org/10.7717/peerj.10353#supplemental-information.

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
