# Peer review of "Spatial and temporal activity patterns of Golden takin (Budorcas taxicolor bedfordi) recorded by camera trapping"

_PeerJ, doi:10.7717/peerj.10353_

## Round 0.1 · original submission · Major Revisions

Overview
This study used a network of camera traps over a wide area to examine the activity of golden takin in a nature reserve in China. Data collected were used to examine diel and seasonal patterns of activity and changes in altitudinal distribution.

The comments of Reviewer 1 are contained in an attached pdf. Both reviewers indicate that the study is potentially publishable but requires major revision to incorporate more appropriate statistical analyses to support and extend its conclusions. Reviewer 1 recommends adding human disturbance as a variable, if possible. Reviewer 2 has provided numerous references to support his suggestions for a more rigorous analysis.

I agree with the issues raised by the reviewers and have some additional concerns. The objectives do not fully correspond to the analysis. There is little or no statistical support for the conclusions of the study. The Discussion needs a more critical analysis, interpretation and integration with the literature. The use of English is quite good overall, but there are numerous small errors of grammar, spelling and word choice. I have indicated some of these by highlights on the attached pdf of the manuscript but have not corrected them because the manuscript is likely to change substantially. It is critical that you have a native English speaker carefully read the entire manuscript to correct the errors.

Major concerns

1. Knowledge gap and study objectives
The Introduction indicates that previous research on this species has not integrated spatial and temporal variation and that the goals of the present research include addressing this gap. It is important to explain what is meant by integration of spatial and temporal patterns, examples of where this has been successfully achieved by previous studies and how the present study will specifically address this. The unresolved questions mentioned on L97-98 have not been explained. My reading of the manuscript finds data on diel and seasonal activity but no information on spatial patterns except Fig. 2, which is not discussed, and changes in altitude which are not related specifically to location. Is it possible to include habitat type (e.g. open grasslands, forest, bamboo, alpine meadow) and human disturbance as variables?

2. Statistical analyses
The reviewers are more expert than I in this research area and provide good introductions to more appropriate statistical analyses. It is notable that the majority of conclusions are not statistically supported and that confidence intervals are not shown for most of the data. Even for temporal patterns, the analyses do not take into account important variation. For example, the diel pattern does not consider seasonal changes in the time of sunrise and sunset.

When introducing the measures of activity (CR, DAI on L142-153) and summarizing the data analysis, it is important to state how you aggregated the data. For example, when calculating spring activity between 12:00 and 14:00, did you divide all independent captures from all locations and years during this season by the number of all independent captures from all locations and years in this season? Many alternative ways of grouping the data are possible. For example you could have calculated this measure separately for each location and each day and then averaged them or you could have grouped data by year or location before calculating the measure. This would have an important effect on the variability of your measures and hence influence the statistics. It is important to have a measure of the variability of the data and a logical decision about grouping the data before estimating variability.

Capture rate is an indirect measure of activity, not a direct measure. In what ways may actual activity patterns differ from the patterns of capture rate? For example, is it possible that for certain types of activity (predator avoidance? human avoidance? mating?) animals are active but do not pass the locations of cameras at water holes or paths?

Daily activity
The presentation of daily activity was not clear. The description of the pattern does not consistently correspond to Fig. 4. This may be a consequence of how the data were graphed. Fig. 4 shows the values of percent activity for each radius line, but these lines are at the beginnings and ends of the measured intervals. The text states that 06:00-08:00 is one interval (L151), but the reader of Fig. 4 does not know if the value from this interval is the one on the 6 or 8 radius line. Would it make more sense to put the average of the 06:00 to 08:00 interval on the 7 line?
Why is winter shown on a different scale from the other seasons in Fig. 4?
Are your data really so precise that providing percentages to two decimal places is meaningful?
Why do the Results on L180ff combine different seasons without considering the differences in daylight period? Would it be helpful to use light shading to indicate mean sunrise and sunset times for each of these seasons? Is it meaningful to provide an average for the whole year despite the day length changes?
On L182 the text refers to ‘a few takin’ without explaining how it was determined that the data represent a few individuals rather than a low level of activity in the population as a whole.

Migration
The only support for the cycle of migration is a chi square that apparently shows differences by month. Since chi square is normally used for categorical data, I don’t see how it was used to test for differences in altitude which is a continuous variable. Furthermore, there is no statistical support for the stated months of maximal and minimal altitude (i.e. whether they differ from adjacent months). No statistics support the stated differences between altitude during the day and night. Also, this analysis does not take into account seasonal differences in sunrise and sunset.

Seasonal patterns of activity
This section raises concerns similar to the above. I do not see how the present analyses could test the effect of temperature on annual activity as implied (L232).

3. Discussion
It is important to carefully distinguish the actual findings of the study (these should be in the Results) from inferences drawn from these findings (these should be in the Discussion).

The Discussion should start with a careful analysis of the confidence that the reader should place in the actual findings. A general statement about limitations of camera traps (L216-221) in insufficient. The Discussion must consider how these and any other limitations of camera traps might have affected the reliability of each of the results. Then discuss what can be inferred from the results and how they relate to the biology of the species. Next, address the originality of these findings. Which findings support or contradict or add new insights to previous research on this topic. Finally, what, if any, general implications can be drawn that might apply more generally to other species in different systems?

The present Discussion begins (L203-224) with much general material that partially overlaps the Introduction and does not relate directly to the results of the research. This needs to be removed or greatly reduced.

The paragraph starting on L236 links findings of this study with previous research to a limited extent, but it is very hard to understand how the two studies are related.

The material presented on L268-273 should be presented earlier when during the discussion of the reliability of the data.

Inferring temperature preferences from data on temperature at the time of camera rcording (L176-178) require information on available temperature as well as the temperature used. Furthermore, it is necessary to consider other trade-offs before drawing conclusions about preferences. For example, animals might leave shaded areas in the summer to forage in areas that are hotter than they prefer because that is where food is available. Thus, additional information is required in Results (available temperatures) and additional discussion is needed to establish the inferences of preference.

L232-235. This is results not discussion.

As noted by the reviewer, it is very important that suggestions for conservation relate directly to your findings. It is not appropriate to list general advice that was already known before this study was carried out. Advice should be derived from the specific findings of this study. For example, how does information on altitude change relate to the need to protect lower elevation habitats? What are the specific altitude ranges of ‘low elevation’ that must be protected? What results lead to the recommendation to reduce human disturbance?


4. Organization of the manuscript
The Introduction needs a much tighter focus on the questions actually addressed by the paper. There is no need for a broad review of the role of behavior in conservation L41-59 because this manuscript does not contribute to this general field except by providing an example from another species. If the authors believe that their study has implications for the field in general, they should indicate these contributions in the Abstract and develop their arguments in the Discussion. If the study contributes to an integration of spatial and temporal patterns, more attention in the Introduction should be focussed on what these questions are and why treating them separately is not appropriate.

Because the goals refer to diel and seasonal activity patterns, the Introduction should carefully define these terms and explain their importance, including insights from important studies of these topics.

L60-81. These two paragraphs are too long and do not provide a well-organized, logical argument for the use of camera traps. Please combine, condense and organize.

The Introduction should provide a short but to-the-point background on what is known of takin biology and behavior that is relevant to the present study. I imagine that diet, breeding season, social organization, habitat and mortality threats could be relevant for a reader of this article. More detailed information about what is already known and what is not known should be provided to establish the knowledge gap addressed by this study. Lines 91-93 list previously studied topics but do not mention their findings.

Other suggestions

Please follow PeerJ Instructions to authors to avoid right-justified margins.
L22-24, 33-34, 35-36. Remove general information and focus on the question asked, answer and implications.
L25. Provide scientific name in Abstract as well as when first used in the main text.
L27 and elsewhere. Use a comma only (no space) to separate thousands from hundreds.
L34-35. As noted by the reviewer, these recommendations are not supported by your study. Replace with specific contributions from this study to biology and conservation of golden takin.
L37. Add family and subfamily to key words.
L37. Behaviour/behavior is spelled inconsistently. Please decide whether you are using UK or US spelling and check the entire manuscript with spell check for consistency of this word and others that vary between the UK and the US.
L50, 52. These seem to be poor choices of references, using papers on horse and antelope activity patterns to describe the discipline of animal behavior. It is much more appropriate cite a major review or textbook synthesis.
L52. This statement appears to contradict L47 in which it was stated that behavior was largely ignored.
L66. Avoid run-on sentences. Start a new sentence with ‘thus’.
L95-98. Condense to reduce redundancy.
L99, 100. Sometimes you refer to ‘golden takin’ and sometimes to ‘takin’. If one term refers to the subspecies you studied and the other to the species in general, please check to confirm that you are using the terms consistently. It would reduce wordiness if you could use ‘takin’ throughout the main manuscript unless it might generate confusion with regard to which subspecies is under discussion.
L133. The SI abbreviation for seconds is s.
L134. Hyphenate when measures are used as adjectives: ‘2-min delay’.
L144. Provide units for all measures. I believe that CR would be ‘independent detections/100 camera days’.
L146. Define ‘effective camera days’.
L149. Units for DAI (%)
L157. ‘Activity intensity’ has not been defined.
L159. Standard deviation is the most appropriate measure for descriptive data, but you used the abbreviation for standard error. Please use standard deviation and correct this statement.
L162-166. The information on sampling effort and success is more appropriate for Methods. Moving it there to reduce redundancy.
L166-170. This information indicates results but not survey effort. It should be incorporated into a separate section or included with the next sections. You provide very little commentary on the spatial distribution reflected in Fig. 2. I think that this is important. Would including elevation and possibly other important variables such as roads or vegetation on the maps make this more useful to readers?
L172. This is redundant because the information is repeated in more detail later in the paragraph.
L173 and elsewhere. You should cite the figures when you first expect the reader to refer to them, not simply at the end of the paragraph. If you cite them once near the beginning of a paragraph, you may not have to cite them multiple times if it is clear from the context that you are still referring to the same figure but if there is any doubt, it is safer to mention the figure, each time the reader needs to check it to verify the information provided.
L176-178. Interpreting the temperature at the time of photographs as the preferred temperature is an inference that requires a logical argument in the Discussion. The result is simply the temperature data. It would be useful to provide the range in addition to SD.
L190 and elsewhere. Use a space before and after symbols such as = and +/-.
L191 and elsewhere. Is a precision of 1 cm in altitude really appropriate or relevant? I suggest that a precision of 1 m is sufficient.
L199. ‘Activity range size’ is a concept that is unclear to me and has not, I believe, been introduced. Please define.
L221. This sentence is not clear.
L265-267. This sentence is irrelevant to your discussion because your goals were not related to migration mechanisms.
L369-374. These references are out of order. Please check all references to confirm that they are in proper alphabetical order.

Figures
In general, the figure captions are insufficient to allow the figures to be understood without reference to the text. Each caption should fully describe the figures. It is not sufficient to have a key that is not mentioned in the caption. Please check some articles in established journals for examples of proper complete captions. Panels should be identified by capital letters as described in PeerJ Instructions to Authors.

Fig. 1. For international readers, the inset map would by more helpful if it showed the location of the reserve in China rather than the Qinling Mountains. Latitude and longitude labels need to be larger in both the main figure and the inset so that they can be easily read. There is a spelling error in the key.

Fig. 2. Activity range has not been defined. There is a spelling error in the caption. In the key, detection should be plural (detections).

Fig. 3. Where was the temperature reading taken? What are the numbers above the bars? Label missing for temperature axis. What are the units on the activity graph? Incomplete frames on panels.

Fig. 4. No explanation for different scale on winter graph. Incomplete frame on panel.

Fig. 5. No scale for left panel. Symbol error in caption. Unreasonable precision (1 cm) for elevation. Season should be below the x-axis in the right panel instead of beside it. The present position forces the graph to be smaller than it needs to be on the page.

Supplementary material
The supplementary data file needs an inset to clearly explain each column and the units. Why are columns F and G included since they are identical throughout the file?

Reviewer 1 ·

Basic reporting

no comment

Experimental design

Please see my comments related to (1) space and time, and (2) detection error in the attached PDF.

Validity of the findings

Please see my comments related to context and validity

Annotated reviews are not available for download in order to protect the identity of reviewers who chose to remain anonymous.

·

Basic reporting

See general comments

Experimental design

See general comments

Validity of the findings

See general comments

Additional comments

The presented manuscript is an original study in the field of studying the temporary activity of animals - an important topic in ecology and wildlife conservation. Understanding the patterns of activity helps not only to better know the ecology and adaptations of a particular species or community, but also to plan measures for its protection or management.

To date, camera traps are a well-known method of non-invasive wildlife research. The methodology of working with them is well studied and accessible to a wide range of specialists. One of the simplest tasks of research (and state variables) with camera traps is the analysis of temporary (daily, seasonal) activity.

It is worth saying that, in general, the level of the presented manuscript doesn’t comply with the international standard of work in this area. The tasks posed by the authors are very simple, and the undertaken analytical approaches are not the most suitable. I believe that the manuscript should be thoroughly redone and ready to provide any help and support to the authors, as I have some experience in carrying out such studies. The following are specific recommendations for improving the manuscript:

Introduction
Chapter Introduction is written very well! The main works are indicated both on the research problems (animal behavior) and on the state of the object of study (golden takin). Nevertheless, little attention is paid directly to the question of studying temporal activity. So, when listing various directions in the study of animal behavior using camera traps for temporary activity, only one link is indicated (Xue et al., 2015). This is clearly not enough, given that it is the temporal activity that the authors pay the most attention to. It is worth noting that the literature on this issue has not been worked out by the authors. I would recommend using the following sources:

Bu H., Wang F., McShea W.J., Lu Z., Wang D., Li S. 2016. Spatial co-occurrence and activity patterns of mesocarnivores in the temperate forests of southwest China. PLoS ONE 11(10): e0164271. DOI: 10.1371/journal.pone.0164271

Di Bitetti M.S., Paviolo A., De Angelo C. 2006. Density, habitat use and activity patterns of ocelots (Leopardus pardalis) in the Atlantic Forest of Misiones, Argentina. Journal of Zoology 270(1): 153–163. DOI: 10.1111/j.1469-7998.2006.00102.x

Foster V.C., Sarmento P., Sollmann R., Tôrres N., Jácomo A.T.A., Negrões N., Fonseca C., Silveira L. 2013. Jaguar and puma activity patterns and predator-prey interactions in four Brazilian biomes. Biotropica 45(3): 373–379. DOI: 10.1111/btp.12021

Gerber B.D., Karpanty S.M., Randrianantenaina J. 2012. Activity patterns of carnivores in the rain forests of Madagascar: implications for species coexistence. Journal of Mammalogy 93(3): 667–676. DOI: 10.1644/11-mamm-a-265.1

Hernández-SaintMartín A.D., Rosas-Rosas O.C., Palacio- Núñez J., Tarango-Arámbula L.A., Clemente-Sánchez F., Hoogesteijn A.L. 2013. Activity patterns of jaguar, puma and their potential prey in San Luis Potosí, Mexico. Acta Zoológica Mexicana 29(3): 520–533.

Linkie M., Ridout M.S. 2011. Assessing tiger-prey interactions in Sumatran rainforests. Journal of Zoology 284(3): 224–229. DOI: 10.1111/j.1469-7998.2011.00801.x

Meek P.D., Ballard G., Claridge A., Kays R., Moseby K., O’Brien T., O’Connell A., Sanderson J., Swann D., Tobler M., Townsend S. 2014. Recommended guiding principles for reporting on camera trapping research. Biodiversity and Conservation 23(9): 2321–2343. DOI: 10.1007/s10531-014-0712-8

Ogurtsov S.S., Zheltukhin A.S., Kotlov I.P. Daily activity patterns of large and medium-sized mammals based on camera traps data in the Central Forest Nature Reserve, Valdai Upland, Russia // Nature Conservation Research. 2018. 3(2). PP. 68–88. DOI: 10.24189/ncr.2018.031

Porfirio G., Foster V.C., Fonseca C., Sarmento P. 2016. Activity patterns of ocelots and their potential prey in the Brazilian Pantanal. Mammalian Biology-Zeitschrift für Säugetierkunde 81(5): 511–517. DOI: 10.1016/j. mambio.2016.06.006

Ridout M.S., Linkie M. 2009. Estimating overlap of daily activity patterns from camera trap data. Journal of Agricultural, Biological, and Environmental Statistics 14(3): 322–337. DOI: 10.1198/jabes.2009.08038

Wearn O.R. & Glover-Kapfer P. 2017. Camera-trapping for conservation: a guide to best-practices. WWF Conservation Technology Series 1(1). WWF-UK, Woking, United Kingdom. 180 p.


Material and methods

In my opinion, this chapter contains quite a few omissions and methodological inaccuracies. Authors should pay attention to how they divide the day into periods. It is necessary to study in detail the statistical analysis of circular data and carefully approach the description of all the software used. At the same time, I want to note that the technical part about camera traps is written quite fully and well and generally meets the requirements (Meek et al., 2014).

Ln. 123. What is GIS 10.1? Perhaps the authors meant ArcGIS?

Ln. 125. It should be indicated how and why such sample design was chosen. For example, provide a link to Wearn O.R. & Glover-Kapfer P. 2017. (P. 80-82).

Ln. 134. Indicate which type of PIR flash was used. Low-glow, no-glow?

Ln. 144. Perhaps it is better to use the more established and common term RAI (Relative Abundance Index) instead of CR? In order not to produce a lot of synonymous terms.

Ln. 156. Why don’t you take into account the twilight period? And why day and night don’t change throughout the year? Where did this interval come from (18:00 - 6:00)?

Ln. 158. Standard deviation is SD, and SE is standard error.

Where did the authors build their plots? Also in SPSS? In what software did the authors process the photos themselves, how did they extract the metadata in Excel spreadsheet?

Ln. 154-159. Authors should study the analytical approaches adopted as standards for analyzing the temporal activity of animals from camera traps. In particular, circular statistics. To compare activity rhythms and find out what activity gravitates to a species, it is better to use not Chi-square and DAI, but special tests: Watson-Wheeler (Lund et al., 2017) and Wald's statistics (Rowcliffe, 2016). For more information see package guides for R:

Lund U., Agostinelli C., Arai H., Gagliardi A., Portugues E.G., Giunchi D., Irisson J.O., Pocernich M., Rotolo F. 2017. Circular statistics. R package version 0.4–93. Available from: https://cran.r-project.org/package=circular

Niedballa J., Courtiol A., Sollmann R., Mathai J., Wong S.T., Nguyen A.T.T., Mohamed A., Tilker A., Wilting A. 2016. Camera trap data management and preparation of occupancy and spatial capture-recapture analyses. R package version 0.99.9. Available from: https://cran.r-project.org/package=camtrapR

Rowcliffe M. 2016. Animal Activity Statistics. . R package version 1.1 https://cran.r-project.org/package=activity

The authors could use then the Manly's selectivity index to observe if the species’ activity can be categorized as diurnal, nocturnal, or crepuscular (see methods in: Bu et al., 2016 or Gerber et al., 2012).

Ln. 171. It is better to call this seasonal activity, rather than annual activity. After all, here we are talking about activity during the year through the seasons, while annual activity implies generalized data on activity during the year (and it is not clear which. Diel?).

Ln. 183. You mention the dawn and the sunset, although you didn’t say anything about these periods in the MatMethods chapter.

You didn’t say anything about the significance of differences in activity between the seasons of the year. Why wasn’t such an analysis conducted? At least Kruskal-Wallis or pairwise multiple comparisons? The same could be done for daily activity in order to statistically identify activity peaks.

You should also clearly understand what you mean by activity. Since the material was collected using camera traps, and the devices themselves were placed along the trails, you can only talk about the activity of movements, but not about the general activity of the takins. These explanations should be given in the M&M chapter.

In the final part of data analysis section, you should report briefly what your null hypothesis is and mention the statistical analyses you will perform.

The chapter Results is well and structured clearly. There are no special questions for it.

Discussion

In general, everything is written very well. Corresponding literature is provided. The research issues are also disclosed. Conclusions are based on the results with a few exceptions.

Ln. 232. "We found no evidence of a significant direct effect of ambient temperature on the annual activity pattern of the species." Sorry, how did you check this and where is the proof of this in the Results chapter?

Ln. 238. "Golden takin exhibited with the most activity at dawn (6:00 to 08:00 am) and dusk (16:00 to 18:00 pm)." You again mention the periods of dawn and dusk, but don’t talk about them in the MatMethods and how they were calculated.

Ln. 291. May be collecting bamboo? Not collecing.

FIG. 1. On the inset map, it would be nice to show in which part of China these mountains are located. It is not clear which region is shown on the map.

FIG. 2. Maps are better represented not as a variation of independent registrations, but as a variation of RAI.

I'm not sure if the authors relate to the work “Activity pattern study of Asiatic black bear (Ursus thibetanus) in the Qinling Mountains, China by using infrared camera traps” for Integrative Zoology journal? In any case, my comments will be the same as for that manuscript. Most likely, the authors have already seen them. But just in case, I repeated some of them.

My common wish for the future is to use the R program for analysis, where a whole range of packages have been developed specifically for analyzing temporary activity and circular data. These are the camtrapR, overlap, activity, and circular packages. With the help of them, the authors will be able to carry out the analysis and present the results more clearly, deeper and more reliably.

---

## Round 0.2 · Major Revisions

Changes in the manuscript have improved the revised manuscript. Reviewer 2 indicates that the manuscript is now acceptable. However, Reviewer 1 and I strongly disagree. The manuscript still requires major changes. It was very disappointing to find that many of the problems that remain were noted in the original comments by the editor and reviewers. Some of these were not adequately addressed in your responses. It is very important that you understand all suggestions made and address them completely. You do not have to agree with them, but you have to have a complete and specific response either showing exactly how you complied or why you disagree. When authors provide incomplete responses, much more work is required of the editor and reviewer because they need to return to issues that they previously identified. Reviewer 1 has been extremely generous in providing detailed appropriate suggestions in an attached pdf. Your rebuttal must include a response to each of these suggestions.

Major gaps in the response to previous suggestions from the editor. Editor’s previous comments in italics. Author’s response in red. Editor’s new comments in regular font. To make it easier to read, I will include these same comments as a pdf because the italics and red font do not show in the formatted response letter from the Editor.

1) It is critical that you have a native English speaker carefully read the entire manuscript to correct the errors. You failed to include this comment in your rebuttal and provided no indication that a native English speaker had checked the manuscript. There remain too many errors in spelling and grammar for me to correct. Since all authors must agree to the submission of the manuscript to PeerJ, it is disappointing to note that even the English speaking author agreed to submit a manuscript with so many mistakes.

2) Knowledge gap and study objectives. The Introduction indicates that previous research on this species has not integrated spatial and temporal variation and that the goals of the present research include addressing this gap. It is important to explain what is meant by integration of spatial and temporal patterns, examples of where this has been successfully achieved by previous studies and how the present study will specifically address this. The unresolved questions mentioned on L97-98 have not been explained. My reading of the manuscript finds data on diel and seasonal activity but no information on spatial patterns except Fig. 2, which is not discussed, and changes in altitude which are not related specifically to location. Is it possible to include habitat type (e.g. open grasslands, forest, bamboo, alpine meadow) and human disturbance as variables?
Answer: Here we added the elevation and vegetation in Figure 1, and spatial analysis.
This answer did not mention a response the request to introduce integration of spatial and temporal patterns, explain the unresolved questions, how you addressed spatial analysis or whether you could include habitat type and disturbance as variables.

3) Statistical analyses. The reviewers are more expert than I in this research area and provide good introductions to more appropriate statistical analyses. It is notable that the majority of conclusions are not statistically supported and that confidence intervals are not shown for most of the data. Even for temporal patterns, the analyses do not take into account important variation. For example, the diel pattern does not consider seasonal changes in the time of sunrise and sunset.
Answer: Temporal analysis- We used Capture Rate (CR) to estimate annual activity pattern of golden takins (Li et al., 2010; Blake et al., 2014). All independent detections for golden takins were summed for each month, and multiplied by 100, and divided by the total number of effective camera-days for each month, CR= No. of independent detections * 100 / No. of effective camera-days.
All photos taken by the cameras recorded date and time, and we estimated daily activity patterns for four periods: dawn (06:00-08:00), dusk (16:00-18:00), the day (08:00-16:00) and night (18:00-06:00). We used a Daily Activity Index (DAI%) of 2-h durations to examine the daily activity patterns: DAI %= No. of independent detections with a duration / Total no. independent detections (Zhang, Bao, Wang, Fang, & Ye, 2012; Liu et al., 2013).
This answer did not address the request to provide statistical support including confidence intervals or to consider seasonal changes in the time of sunrise and sunset.

4) When introducing the measures of activity (CR, DAI on L142-153) and summarizing the data analysis, it is important to state how you aggregated the data. For example, when calculating spring activity between 12:00 and 14:00, did you divide all independent captures from all locations and years during this season by the number of all independent captures from all locations and years in this season? Many alternative ways of grouping the data are possible. For example you could have calculated this measure separately for each location and each day and then averaged them or you could have grouped data by year or location before calculating the measure. This would have an important effect on the variability of your measures and hence influence the statistics. It is important to have a measure of the variability of the data and a logical decision about grouping the data before estimating variability.
Answer: Detail please see quesiton 2. And we have increased Talbe 1 to detaile introduction.
This answer did not address the need to explain how you aggregated the data.

5) Capture rate is an indirect measure of activity, not a direct measure. In what ways may actual activity patterns differ from the patterns of capture rate? For example, is it possible that for certain types of activity (predator avoidance? human avoidance? mating?) animals are active but do not pass the locations of cameras at water holes or paths?
Answer: We chose “Capture Rate” as measure of activity based on: 1) annunal activity patterns of our results reveal that there was good accordance with the known activity patterns of species. 2) capture rate consider the relations of No. of independent photographs and No. of Effective camera days. 3) this method was also widely accepted in published paper.
Our results only consider the moving behavior of golden takins. The suggestion put forward by the reviewers is the focus of our next work. Now we have investigated clearly the distribution of golden takins, next we will focus the infrared cameras on the water holes, mating area to study other activities. Especially our results found the golden takins have been frequently reported to forage and damage local agricultural crops (e.g., wheat, lettuce) in low-elevation areas, we will investigate which crops the golden takins most like to eat, and guide local residents to change their crops.
Golden takins have only few natural enemies, our study systematically surveyed from April 2014 to October 2017, but only one photograph of leopard in study area.
Your manuscript is a study of activity patterns. This question requested that you consider how capture rate by cameras might not provide a completely accurate measure of activity pattern. Your answer did not address this. Although camera traps have been used to study activity patterns in other studies, it is important for you and readers to understand potential biases in this measure in your study.

6) Why do the Results on L180ff combine different seasons without considering the differences in daylight period? Would it be helpful to use light shading to indicate mean sunrise and sunset times for each of these seasons? Is it meaningful to provide an average for the whole year despite the day length changes?
Answer: We just based on the time of photographs, not on the shading. It is difficult to judge shading happen in night, dawn or dusk.
You did not explain why it was valid to combine different seasons without considering the difference in the daylight period. You misunderstood the comment about shading. I was suggesting that the graph of activity pattern could use shading (light gray areas) to indicate dawn and dusk as in many publications on activity periods.
7) On L182 the text refers to ‘a few takin’ without explaining how it was determined that the data represent a few individuals rather than a low level of activity in the population as a whole.
Answer: The individuals in populations or just only individual. Golden takins sometimes got together and sometimes separated, we have no idea to judge population or individual.
This answer says that you could not separate individuals. I understand that, which is why I questioned your statement about a few takin. Since you don’t know how many takin were active during that period, the statement ‘a few takin’ is not supported by your data. You continued to use similar, unjustified terms in the revised manuscript (L183).

8) The only support for the cycle of migration is a chi square that apparently shows differences by month. Since chi square is normally used for categorical data, I don’t see how it was used to test for differences in altitude which is a continuous variable. Furthermore, there is no statistical support for the stated months of maximal and minimal altitude (i.e. whether they differ from adjacent months). No statistics support the stated differences between altitude during the day and night. Also, this analysis does not take into account seasonal differences in sunrise and sunset.
Answer: Here we have deleted the Chi-square statistic, and used Kruskal-Wallis to test difference in monthly elevation. We calculated the monthly average elevation ranges of camera trapping detected locations of species to describe the seasonal migration changes. The result of seasonal migration described (Figure 3):
Although you changed the analysis, you still did not provide statistical support for differences between different months or respond to the comment about differences between day and night. It might be useful to consult with a statistician regarding all your statistical analyses.
9) Because the goals refer to diel and seasonal activity patterns, the Introduction should carefully define these terms and explain their importance, including insights from important studies of these topics.
Answer: In introduction part, we add the sentence to focus activity patterns: Activity patterns are fundamentals aspects of animal behavior and important in determining the distribution of individuals in space and time, and an increasing number of researches are using camera traps to survey activity patterns of animals (Gerber et al., 2012; Ikeda et al., 2015; Xue et al., 2015; Bu et al., 2016; Frey et al., 2017; Blake and Loiselle, 2018; Bohm and Hofer, 2018).
Your answer does not provide a true definition daily and seasonal activity patterns. It states their importance but does not explain why. It provides references but not what insights they provided. I am not asking for a much longer Introduction, but one that provides insights into the biological importance of these patterns.

10) The Introduction should provide a short but to-the-point background on what is known of takin biology and behavior that is relevant to the present study. I imagine that diet, breeding season, social organization, habitat and mortality threats could be relevant for a reader of this article. More detailed information about what is already known and what is not known should be provided to establish the knowledge gap addressed by this study. Lines 91-93 list previously studied topics but do not mention their findings.
Answer: we added the paragraph in discussion part to introduce the implications of our findings, and the difference with previously studies.
The paragraph you added summarizes the topics addressed by previous literature but not what was found. The reader should have a concise summary of the findings of previous studies that are relevant to the findings of this study.
11) Other suggestions
Answer: we have modified the suggestions clearly new revised version.
My comments included more than two pages of specific comments. You need to respond individually to each of these comments. These were not merely small issues of grammar or spelling but important suggestions to clarify the manuscript. Skipping these was particularly disappointing because some of the problems previously identified in the revised manuscript.
Additional Editor comments:

L44. Inappropriate to use a book review to support a major generalization.
L50. Use U.S. or British spelling conventions consistently throughout the manuscript. You can do this using spell check with the appropriate language choice.
L90. Seasonal and annual activity patterns have not been defined. I think that for many readers these terms would mean the same thing.
L108. I don’t think that ‘weeds’ is a proper botanical term. What do you mean?
L173. What is the evidence for two migrations per year?
L196-203. This paragraph is about the originality of your approach rather than about what you found. The need for your approach belongs in the Introduction.
L268-272. The present Conclusions section is too vague and general. Please check the Instructions to Authors for PeerJ. Conclusions is a section in which you indicate unresolved questions, gaps and future directions.
References: check all references carefully. They are quite accurate compared to some other manuscripts that I have seen but there are cases of capital letters missing from where they belong (book titles) and present where they don’t belong (article titles), missing italics from species names and spelling errors (L403).
Table 1. Heading is incomplete (missing units for capture rate) and incorrect style (we summarized). Descriptive data should use SD not SE. Is the sample size for elevation the same as number of independent photographs?
Fig. 3. Explain error bars in caption. Inserted key is not needed.
Fig. 4. Define DAI in caption.
Fig. 5. Provide units for CR.

Reviewer 1 ·

Basic reporting

I have recommended some English language the authors are welcome to use or adapt. I would repeat the editor’s comment in the initial decision that “it is critical that you have a native English speaker carefully read the entire manuscript to correct the errors.”

Experimental design

The study design (L116 to L131) is unclear and needs to be rewritten for clarity and repeatability. I have outlined some key points in my response that I believe need to be addressed.

Validity of the findings

The authors have dramatically improved their results and discussion. They are, however, not reporting the Kruskal-Wallis test correctly. The KW only indicates that at least one sample dominates other samples. If they wish to compare multiple samples (as they do in the text) they need to conduct a post hoc test for the specific pairs. The most common approach is to use Dunn’s test.

I outline this in my review.

Additional comments

Your revisions have improved the manuscript - I appreciate the work that you put into revising this draft. The methods, results, and discussion are much more aligned with their objectives and each other. The manuscript still requires revisions particularly around the study design.

Please see my detailed review (PDF).

Annotated reviews are not available for download in order to protect the identity of reviewers who chose to remain anonymous.

·

Basic reporting

See General comments

Experimental design

See General comments

Validity of the findings

See General comments

Additional comments

In general, I was pleased with the corrections made by the authors. They took into account most of my comments. Those comments that they did not take into account, they justified and explained the reason. The manuscript underwent significant changes after the review, which, in my opinion, only benefited it. Many thanks to the editor and second reviewer for valuable comments and additions that undoubtedly improved the manuscript. I wish the authors to make the latest corrections and prepare the final version of their manuscript. I hope that in their subsequent articles they will be able to realize all their plans!

---

## Round 0.3 · Minor Revisions

I invited only Reviewer 1 to look at this revision of your manuscript because Reviewer 2 had found the previous version acceptable. I am pleased that Reviewer 1 was available and provided a rapid and very thorough review with suggestions that will make your manuscript clearer to readers. Note that most of the reviewer’s comments are provided in an attached pdf. You should respond to each of these comments when your submit your revision. I have some additional comments of my own and have provided a number of language corrections on an attached pdf. You should respond to my comments below, but you do not need to respond to the minor suggestions on my pdf unless you disagree with them.

Your revision has very substantially improved the manuscript and it is now close to ready for publication. However, a few major issues and many minor ones remain.

Reviewer 1 points out (#1) that your sampling design is not strictly random with regard to location. I agree that this is true but do not think that is serious enough to reject the manuscript. I agree with his suggested clarification of the methods.

Reviewer 1 points out (#2) that you should provide the units for capture rate as ‘number of detections per 100 trap days’. I agree.

Reviewer points out (#3) that your statistical analysis does not support two migrations per year. Such a migration is also not clearly evident in Fig. 4. I suggest that your Results and Abstract should state support for only one migration. It would be acceptable to suggest that the small changes in elevation between December and February suggest a possible migration during this period but that further research is required to determine whether this is a true pattern.

I agree with the Reviewer’s point (#4) that you should clarify the issue of detection probability when a takin is present and indicate any procedures that you took to increase detection.

I have checked the reviewer’s other specific comments and find them appropriate and very helpful. I am grateful for the careful work of this reviewer. In a few cases, I have suggested alternative wording in my comments on the pdf. With regard to L192, I am willing to accept the utilization comparisons without statistical support, as long as you use language, as suggested by the reviewer, that is clear that your interpretation is based on visual inspection of the data rather than statistical analysis.

It is necessary to avoid or reduce as much as possible redundancy of information in the text, tables and figures. Some of your tables and figures overlap extensively. Below, I have suggested which ones to eliminate for a more concise manuscript.

Final version of the manuscript. The reviewer noted some discrepancies between the track changes and submitted pdf version of the manuscript. Before this reviewer began the detailed review, he emailed PeerJ to ask which version should be followed. I looked at the two versions and found some additional discrepancies. I considered sending the manuscript back to you to resolve the differences but decided that the number of such errors was fairly small and not worth the additional delay. However, the mistake cost the reviewer and me extra time unnecessarily and could have caused a significantly delay for you and your co-authors. In the future, please be careful that both versions match completely. When you make your changes following our grammatical suggestions, please be certain that you are working on the correct version of the manuscript.

Additional suggestions
L193 and elsewhere. The U.S. spelling is utilization. Please use it consistently. The reviewer noted the problem but spelled it the same way twice in the pdf (perhaps it was ‘corrected’ by his spelling checker).

Table 1. I agree with reviewer’s suggested change in heading. Add comma to numbers of 1,000 and greater. P is never zero. Change to P<0.001.

Table 2. I agree with reviewer’s suggestion to remove table. Redundancy of data between tables, figures and text should be avoided as much as possible.

Table 3. I agree with reviewer’s suggested change in heading. Put a space between the number and parenthesis at the start of the percent [e.g., 2 (0.18%), not 2(0.18%)]

Table 4. I agree with reviewer’s suggested change in heading. Add comma to numbers of 1,000 and greater. Change to P<0.001.

Table 5. Remove Table 5 because it is redundant to Fig. 5.

Fig. 4. Remove Fig. 4 because it is redundant to Table 4, which provides more information. However, the use of SE in the table and quartiles in the figure is ambiguous. If the data are normally distributed, mean and SD are appropriate. If not normally distributed, data should be described by median and interquartile range.

Fig. 5. Remove the sentence that describes results in the caption. I do not agree with the reviewer’s suggestion to change the form of the figure. I find it acceptable to use the mean values alone if you prefer, although adding SD (if the data are normally distributed) or switching to median and quartiles would (if data are not normally distributed).

Fig. 6. Remove Fig. 6 because it is redundant to Table 1, which provides more information.

Reviewer 1 ·

Basic reporting

The authors have really improved the basic reporting. There are some grammatical and typographic errors that should be addressed before publishing but overall I think it is it fairly good shape.

Experimental design

Please see my PDF comments in relation to the study design.

Validity of the findings

The interpretation of the elevation results is not correct and needs to be updated to reflect the results of the post hoc analysis.

Additional comments

Once again your revisions have improved the manuscript, and I appreciate the work that you put into revising this draft. Please see my detailed review for details (PDF).

Annotated reviews are not available for download in order to protect the identity of reviewers who chose to remain anonymous.

---

## Round 0.4 · Minor Revisions

I apologize for the long delay in reviewing your revised article. I did not think it necessary to send the manuscript out for review again, but I was unable to find the time to complete a review right away. The manuscript is very close to ready for publication, but there are a number of minor errors, mostly changes in wording needed for clarity or grammatical improvements, especially figure captions. See the annotated pdf for details. I am sorry that I missed some of the needed corrections in the previous version. As before, I highlighted sections that needed changes and inserted comments to suggest those changes. If you don’t agree with the changes or are not sure you understand what I mean, contact me directly by email <[email protected]> so that we can decide the best course of action before you resubmit. The corrections should not take you long, and I would like the next version to be completely acceptable. Below, I have a few comments on specific changes to help explain why changes are needed.

L185. I think you mean Fig. 2 not Fig. 1 here. Please check that all text references to tables and figures are correct.
L224-226. To reduce redundancy, I suggest removing the statistics from the text because they are already in the table. Adding the word ‘significantly’ will alert readers to check the table for the statistics.
L361, 365. Use lower case for words in the titles of articles other than proper names. You did this with the majority of references, but a few titles still have capital letters where they are not needed.
Table 1. I have added text to clarify how the reader should interpret the letters from the Duncan test in the heading and removed the footnote as well as removing some unneeded words from the heading. I think the table will be clearer if the column ‘No. of effective camera days’ becomes the third column (after months).
Table 3. I suggest changes similar to those in Table 1.
Fig. 1. I suggested text to provide a more complete description of the figure. Note the difference in spacing for 2 x 2.
Fig. 2. I suggest changes to make the caption more accurate. The figures show the distributions, not the changes in distribution as the caption stated. The changes are something the reader infers by inspecting the distributions. Note that the caption provided had incorrect information about which season was in which panel. It is important to alert readers to the change in value of dot sizes in different panels.
Fig. 3 is redundant to Table 2. However, I will allow you to keep both in because I did not recognize it in the previous version.
Fig. 4. I suggest additional text for the caption. The y-axis should identify the units in the figure. Daily rhythm is what a reader infers from the figure, not the units. The units are Daily Activity Index (%). The axis does not need labels to a tenth of a percent; remove decimal and zero and percent sign.

---

## Round 0.5 · accepted · Accept

I appreciate the prompt and thorough correction of the manuscript and the clarity of the rebuttal document. The manuscript is now ready for publication. Congratulations on a very nice study.